# Paeoniflorin Inhibits Porcine Circovirus Type 2 Replication by Inhibiting Autophagy and Targeting AKT/mTOR Signaling

**DOI:** 10.3390/vetsci12020117

**Published:** 2025-02-02

**Authors:** Zhengchang Wu, Luchen Yu, Yueqing Hu, Wenbin Bao, Shenglong Wu

**Affiliations:** 1Key Laboratory for Animal Genetics, Breeding, Reproduction and Molecular Design of Jiangsu Province, College of Animal Science and Technology, Yangzhou University, Yangzhou 225009, China; zcwu@yzu.edu.cn (Z.W.); luchenyu202209@163.com (L.Y.); 18595384877@163.com (Y.H.); wbbao@yzu.edu.cn (W.B.); 2International Research Laboratory of Prevention and Control of Important Animal Infectious Diseases and Zoonotic Diseases of Jiangsu Higher Education Institutions, Yangzhou University, Yangzhou 225009, China

**Keywords:** porcine circovirus type 2, paeoniflorin, autophagy, AKT/mTOR signaling

## Abstract

Porcine circovirus type 2 (PCV2) is the main pathogen causing postweaning multisystemic wasting syndrome, which leads to enormous losses for the porcine industry. Due to the instability of commercial vaccines, it is necessary to identify effective antiviral drugs during PCV2 infection. In this study, we focused on the effect of paeoniflorin on PCV2 infection in porcine kidney cells (PK15). We found that paeoniflorin inhibits porcine circovirus type 2 replication by inhibiting autophagy by targeting AKT/mTOR signaling. Our study provides new information that could help the implementation of the antiviral drug paeoniflorin in PCV2 infection in the future.

## 1. Introduction

Porcine circovirus (PCV) belongs to the genus circovirus within the family Circoviridae. It has no envelope and contains covalently closed single-stranded circular negative-sense DNA. Currently, four serotypes of porcine circovirus have been identified, namely PCV1, PCV2, PCV3, and PCV4 [1]. In 1991, a Canadian scholar, Clark, isolated a circovirus from diseased pigs suffering from postweaning multisystemic wasting syndrome (PMWS). Subsequently, Meehan et al. named this pathogenic circovirus porcine circovirus type 2 (PCV2). PCV2 is the smallest single-stranded circular DNA virus known to infect mammals, and the replication of PCV2 is heavily dependent on host cellular mechanisms [2]. PCV2 is also a major pathogen causing dermatitis and nephrotic syndrome, porcine interstitial pneumonia, sow reproductive disorders, infectious congenital tremors in piglets, etc. [3]. Since its discovery in the late 1990s, PCV2 has become a major pathogen causing porcine circovirus disease (PCVD) in pig production worldwide, thus leading to huge economic losses to the global pig industry [4,5,6]. However, the pathogenic mechanism of PCV2 is still unclear, and the main natural transmission route remains oronasal contact [7,8]. Lymphocyte exhaustion and histiocyte replacement in lymphoid tissues are the hallmark lesions of PCV2 infection [9]. PCV2 infection and replication in lymphoid tissues can disrupt the structure of lymphoid follicles, leading to lymphocyte exhaustion and subsequent replacement by histiocytes [10]. Thus, lymphoid follicle destruction and leukopenia associated with PCV2 infection can lead to immunosuppression in pigs. In addition, studies have demonstrated that PCV2 can replicate in bronchial and inguinal lymph nodes, the tonsils, lungs, liver, kidneys, spleen, and thymus tissue [11,12,13,14]. Numerous studies have shown that when PCV2 infects cells, the expression of cellular genes is generally altered, including the regulation of cellular signaling pathways, the induction of oxidative stress, apoptosis, and autophagy, the enhancement of the expression of inflammation-related genes, etc. [2]. Although the widespread use of commercial vaccines has reduced the subclinical symptoms of viral infections, the rate of antigen positivity remains high in the face of mixed infections with different PCV2 genotypes, constant genotypic variation, and different clinical manifestations of infection [15]. However, it is clear that vaccination cannot provide complete protection for the host against PCV2 infection.

Nowadays, a number of drugs with antiviral effects have been explored and developed. Natural products are a rich source of bioactive molecules and bear potential pharmaco-therapeutic functions. It has been shown that natural extracts of scopoletin, artemisinin B, and artemisinic acid (a single fraction isolated from Artemisia annua) can interfere with the viral insertion and replication process by interacting with the 3CLpro and Spike proteins of SARS-CoV-2, which can help to combat the SARS-CoV-2 variant and other possible coronaviruses [16]. It has also been shown that polyphenol-rich sugarcane extract (PRSE), an extract from sugarcane treated with PRSE in influenza a virus (IAV)-infected cells, can inhibit the early stages of infection and reduce viral genome replication, mRNA transcription, and viral protein expression [17]. In addition, PRSE treatment also attenuates the replication of multiple H3N2 and H1N1 subtypes of IAV strains [18]. As a result, various plant extracts and natural components are continuously being exploited for their potential value and become novel antiviral agents. However, there are still relatively few studies on PCV2 at present; this lack of in-depth research has hindered the development of more targeted and efficient strategies to deal with PCV2-related issues.

Paeoniflorin (PF) is a monoterpene glycoside compound, which is the main bioactive component of *Paeonia lactiflora* [19]. During the cultivation of medicinal paeonies, the above-ground parts are often discarded as waste, resulting in a huge waste of resources. It has been recorded that PF, as an important active ingredient in paeonia lactiflora, has the ability to regulate immunity, antioxidant, anti-inflammatory, neuroprotective, inhibit tumor cells, and carry out other pharmacological activities [20,21,22]. Studies have shown that PF also has some therapeutic effects on viral infections. For example, some researchers have investigated the effects of PF on myocardial injury in coxsackie B3 virus (CVB3)-induced viral myocarditis (VMC) mice and its possible mechanism of action. The results showed that PF attenuated the inflammatory response of cardiomyocytes in VMC mice, reduced cardiomyocyte apoptosis, and ameliorated the myocardial tissue injury caused by CVB3 by inhibiting the activation of NLRP3 inflammatory vesicles [23,24,25]. Food rich in PF would be potential health-oriented foods, while natural plant extracts such as PF also have the prospect of applications as functional feed additives for livestock and poultry. As the main therapeutic component of paeonia lactiflora, PF has a wide range of applications and is a drug with great market prospects. However, studies on the effect of paeoniflorin on PCV2 are scarce, and its mechanism of action remains unclear, thus urgently awaiting our exploration.

To elucidate the role and underlying mechanism of PF in PCV2 replication, we use porcine kidney cells (PK15) as a model to investigate the effect of PF on PCV2 infection. This study first confirmed that PF significantly inhibited PCV2 replication in PK-15 cells. We then verified that the AKT/mTOR signaling pathway is closely associated with PCV2 replication. Finally, we investigated the regulation of the AKT/mTOR pathway by PF and elucidated the role of PF in regulating PCV2 replication through the AKT/mTOR pathway. Herein, this study facilitates the use of PF in the treatment of PCV2 infection in the future and provides a basis for further research on the mechanism of PCV2 replication.

## 2. Materials and Methods

### 2.1. Cell Culture, RAPA Treatment, and PCV2 Infection

PK15 cells (ATCC, CCL-33) were cultured in Dulbecco’s modified Eagle’s medium (DMEM) (Thermo, Waltham, MA, USA) supplemented with 10% fetal bovine serum (FBS) (Bio-Channel, Nanjing, China) and penicillin/streptomycin (100 U/mL) (Solarbio, Beijing, China) in a humidified atmosphere of 5% CO_2_ and 95% air at 37 °C. PF (≥98%, HPLC) (DASF, Nanjing, China) was dissolved in DMEM for 48 h. Rapamycin (RAPA) (MedChemExpress, Monmouth Junction, NJ, USA) was dissolved in DMEM to a concentration of 0.1 nM for 48 h. Additionally, PCV2d was preserved in our laboratory. PK15 cells were infected with PCV2d for a multiplicity of infection (MOI) of 1 for all experiments conducted.

### 2.2. CCK-8 Assay

PF was dissolved in the dimethyl sulfoxide (DMSO) (Solarbio, Beijing, China). A total of 20,000 cells (100 µL) per well were uniformly inoculated into 96-well plates containing 10% FBS medium. After incubation in a 37 °C, 5% CO_2_ incubator for 24 h, the medium was replaced with the medium containing different concentrations of PF (15 mM, 30 mM, 60 mM, 125 mM, 250 mM, 500 mM, 1000 mM, 100 mM, 125 mM, 150 mM, 175 mM, 200 mM, 225 mM, 250 mM, 275 mM, 300 mM, 400 mM, 425 mM, 450 mM) in DMEM for 48 h. The Cell Counting Kit-8 (CCK-8) (Vazyme, Nanjing, China) was used to detect cell viability. The CCK-8 solution (Vazyme, Nanjing, China) was added to 96-well plates at 10 µL per well. Optical density values were detected at 450 nm using a Tecan Infinite 200 microplate reader (Sunrise, Tecan, Zurich, Switzerland).

### 2.3. Indirect Immunofluorescence Assay

Cells were infected with PCV2 and collected at different time points for the experiments. Cells were washed with phosphate-buffered saline (PBS), fixed by adding 4% paraformaldehyde, and incubated at 37 °C for 30 min. The cells were then treated with 0.5% Triton X-100 (Vazyme, Nanjing, China) for 15 min to rupture the cell membrane, and then 5% bovine serum albumin (BSA) (Solarbio, Beijing, China) was added and the cells were closed for 2 h at 37 °C. This was followed by incubation with PCV2 capsid (PCV2 *CAP* Antibody, VMRD, Pullman, WA, USA) and secondary Alexa Fluor 555 Conjugated Goat anti-mouse IgG (Huabio, Beijing, China). Finally, 4′,6-diamidino-2-phenylindole (DAPI) was added and incubated for 5 min, and then cells were treated by adding an antifade mounting medium for fluorescence (Biosharp, Beijing, China). The cells that underwent different treatments were then observed under a fluorescence microscope (Leica Microsystems, Wetzlar, Germany).

### 2.4. qPCR Analysis

The RNA extraction was isolated using the TRIzol reagent (Thermo, Waltham, MA, USA), and cDNA was synthesized using the HiScript II Q Select RT SuperMix with the gDNA Remover or miRNA 1st Strand cDNA Synthesis Kit (Vazyme, Nanjing, China) according to the manufacturer’s instructions. The DNA extraction was isolated using the FastPure Cell/Tissue DNA Isolation Mini Kit (Vazyme, Nanjing, China) according to the manufacturer’s instructions. AceQ qPCR SYBR Green Master Mix (Vazyme, Nanjing, China) was used to perform qPCR. Three replicates were set for each sample, and the CT value was average. Based on the gene sequences published in the GenBank database, qPCR primers were designed using Premier 6.0 software with *β-actin* as the internal reference gene (Appendix A). All primers were prepared using TsingKe (TsingKe, Beijing, China).

### 2.5. Western Blot Analysis

After PCV2 infection, cells were collected and washed with PBS, pre-cooled at 4 °C, and a mixture containing the protein lysis solution (Applygen, Beijing, China), 100× phosphatase inhibitor (TargetMOI, Boston, MA, USA), and 100× protease inhibitor (TargetMOI, Boston, MA, USA) was added. After lysing the cells for 20 min at low temperature, the supernatant was obtained by centrifugation at 4 °C, 12000 rpm for 15 min. A bicinchoninic acid kit (Yeason, Shanghai, China) was used to normalize protein levels. A 5 × loading buffer (Vazyme, Nanjing, China) was added and denatured at 98 °C for 10 min. Finally, the denatured samples were stored at −20 °C.

The treated samples were separated using 10% sodium dodecyl sulfate–polyacrylamide gels and electro-transferred onto polyvinylidene fluoride (PVDF) membranes (Millipore, Billerica, MA, USA). After blocking for 2 h at room temperature, primary antibodies (PCV2 *CAP* antibody, GeneTex, San Antonio, TX, USA; HSP90 antibody, Proteintech, Wuhan, China; P62 antibody, Proteintech, Wuhan, China; LC3 antibody, Proteintech, Wuhan, China; mTOR antibody, Abcepta, Hangzhou, China; p-mTOR antibody, Huabio, Hangzhou, China; AKT antibody, Santa Cruz, Shanghai, China; p-AKT antibody, Santa Cruz, Shanghai, China) were incubated overnight at 4 °C. The PVDF membranes were washed by TBST and incubated with secondary antibodies (Proteintech, Wuhan, China) at room temperature for 2 h. The concentration of the primary antibody was 1 µg/mL, and that of the secondary antibody was 0.02 µg/mL. After being washed with TBST, samples were analyzed by ECL development (Tanon, Shanghai, China).

### 2.6. Confocal Immunofluorescence Microscopy

PK-15 cells stably expressing lentiviral pLV-GFP-mCherry-LC3 were inoculated on the slides of 12-well plates. After treatment with PF and RAPA, the cells were inoculated with PCV2 for 2 h and then incubated with 2% FBS DMEM, respectively, for 48 h. After fixation and staining of the nuclei with DAPI, the cells were observed under a confocal laser scanning microscope.

### 2.7. Flow Cytometry Analysis

The Annexin V-FITC/PI Apoptosis Detection Kit (Solarbio, Beijing, China) was adopted to detect changes in the apoptosis levels of PK15 cells and PF-treated cells. The treated cells were digested with trypsin into centrifuge tubes without EDTA and centrifuged to remove the supernatant. The single-stained tube group was used as a positive control, and 5 µL of the Annexin V/FITC mix was added and incubated at room temperature in the dark for 5 min. Solution without Annexin V/FITC and propidium iodide was used as the negative control group. Annexin V/FITC and PI were added to the treatment group. Apoptosis was detected using a FACScan flow cytometer (Becton Dickinson, Franklin Lake, CA, USA). Analysis was performed using CytExpert 2.3 and FlowJo 7.6 software.

### 2.8. RNA Sequencing

After 48 h of PF treatment, PK15 cells were collected from the blank control group (NC, n = 4) and PF treatment group (PF, n = 4). Total RNA was extracted using the Trizol reagent, and the quality of RNA was assessed by 1% formaldehyde denaturing agarose gel electrophoresis. The RNA concentration was determined using an ND-1000 nucleic acid/protein concentration analyzer. Subsequently, all RNA samples were converted into double-stranded cDNA and sequenced on the Illumina Hiseq2500 platform from Oebiotech (Shanghai, China). Differential expression genes (DEGs) were identified based on the principles of corrected *p*-values < 0.05 and |log_2_(foldchange)| < 1. The Gene Ontology (GO) function of DEGs was labeled on the returned data using GOseq. The enrichment of DEGs in the Kyoto Encyclopedia of Genes and Genomes (KEGG) pathway was detected using KOBAS software (version 3.0). GO terms and KEGG pathway analysis were performed using a corrected cut-off value of 0.05. The raw high-throughput RNA-seq data from this analysis were uploaded to the NCBI SRA (Sequence Read Archive) database (accession number PRJNA1173311).

### 2.9. Statistical Analysis

SPSS 18.0 software (Chicago, IL, USA) was employed to conduct statistical analyses. Relative quantitative results were examined via the 2^−ΔΔCt^ method. Results were represented as mean ± standard deviation (SD), followed by a comparison via Student’s *t*-test. *p* < 0.05 or *p* < 0.01 indicated statistical significance. Semi-quantification was performed using ImageJ software (National Institutes of Health, Bethesda, MD, USA).

## 3. Results

### 3.1. Safety Assessment of PF in PK15 Cells

In order to investigate the safe working concentration of PF, we treated PK15 cells with different concentrations of PF (100 mM, 125 mM, 150 mM, 175 mM, 200 mM, 225 mM, 250 mM, 275 mM, 300 mM, 325 mM, 350 mM, 375 mM, 400 mM, and 425 mM, 450 mM) before incubation for 48 h. After that, the effects of different concentrations of PF on the activity of PK15 cells were detected according to the CCK8 method. The results showed that there was no clear toxic response to PK15 cells when the PF concentration was 0–250 mM (Figure 1A); therefore, we further subdivided the concentration gradient, and the results showed that there was no obvious toxic response to PK15 cells when the PF concentration was 0–275 mM; moreover, the addition of PF from 300 mM onwards significantly decreased the cell activity (Figure 1B). Therefore, the three acting concentrations (100 and 200 mM), which did not cause a toxic response to PK15 cells, were taken for subsequent experiments. A blank control and three concentrations of PF-treated groups were set up, and an apoptosis assay was performed after 48 h of drug treatment. The results of flow cytometry showed that there was no significant difference in the apoptosis level of PK15 cells between the drug-treated group and the blank control group (Figure 1C), indicating that PF does not affect the apoptosis level of PK15 cells (Figure 1D). To conclude, when we used PF at concentrations of 0–275 mM, there was no significant inhibition of PK15 cell viability or apoptosis.

In order to investigate the safe working concentration of PF, we treated PK15 cells with different concentrations of PF (100 mM, 125 mM, 150 mM, 175 mM, 200 mM, 225 mM, 250 mM, 275 mM, 300 mM, 325 mM, 350 mM, 375 mM, 400 mM, and 425 mM, 450 mM) before incubation for 48 h. After that, the effects of different concentrations of PF on the activity of PK15 cells were detected according to the CCK8 method. The results showed that there was no clear toxic response to PK15 cells when the PF concentration was 0–250 mM (Figure 1A); therefore, we further subdivided the concentration gradient, and the results showed that there was no obvious toxic response to PK15 cells when the PF concentration was 0–275 mM; moreover, the addition of PF from 300 mM onwards significantly decreased the cell activity (Figure 1B). Therefore, the three acting concentrations (100 and 200 mM), which did not cause a toxic response to PK15 cells, were taken for subsequent experiments. A blank control and three concentrations of PF-treated groups were set up, and an apoptosis assay was performed after 48 h of drug treatment. The results of flow cytometry showed that there was no significant difference in the apoptosis level of PK15 cells between the drug-treated group and the blank control group (Figure 1C), indicating that PF does not affect the apoptosis level of PK15 cells (Figure 1D). To conclude, when we used PF at concentrations of 0–275 mM, there was no significant inhibition of PK15 cell viability or apoptosis.

### 3.2. Construction of PCV2-Infected PK15 Cell Model

In order to detect the proliferation pattern of PCV2 (MOI = 1) on PK15 cells and to determine a time point for the subsequent collection of samples, we used qPCR, Western blot, and the indirect immunofluorescence assay (IFA) to detect the expression of the PCV2 *CAP* gene at different time points. As is shown in Figure 2, after the PCV2 infection of PK15 cells, the replication level of the PCV2 *CAP* gene in PK15 cells from 0 to 48 h showed an increasing trend with time. However, the expression of the PCV2 *CAP* gene in PK15 cells slightly decreased over a 48–72 h period (Figure 2A). Both Western blot (Figure 2B) and IFA (Figure 2C) results were consistent with the quantitative results. Thus, we successfully established a model of PCV2-infecting PK15 cells and determined that 48 h is the peak time point for PCV2 replication.

### 3.3. Effect Analysis of PF on PCV2 Replication

To investigate whether PF inhibits the replication of PCV2 in PK15 cells, qPCR, Western blot, and IFA were employed to detect the expression of the PCV2 *CAP* gene. The experiment was designed with different concentrations (0 mM, 5 mM, 15 mM, 25 mM, 50 mM, 100 mM, 200 mM) and treatment methods (pre-treatment (PF was added 24 h before PCV2 addition), simultaneous treatment (PF was added simultaneously with PCV2), continuous treatment (pre-treatment + simultaneous treatment), and post-treatment (PF was added 24 h after PCV2 addition)), and the conditions at the same time points (24 h, 48 h, 72 h) were compared. In the concentration experiment, the addition of PF reduced the expression of the PCV2 *CAP* gene, suggesting its inhibitory effect on PCV2. Moreover, it was found that the inhibitory effect of paeoniflorin on PCV2 replication was enhanced with the increase in treatment concentration, showing a dose-dependent manner (Figure 3A). The Western blot results were consistent with the quantitative results (Figure 3B and Appendix A). Subsequently, cells infected with PCV2 were treated with the same concentration of PF but with different treatment methods. The results indicate that when the treatment method was not used post-treatment, the expression of the PCV2 CAP gene in the PF-treated group was significantly reduced compared with that in the negative control group (Figure 3C), and the Western blot results were consistent with the quantitative results (Figure 3D and Appendix A). In particular, the PF pre-treatment had the most obvious inhibitory effect on PCV2, suggesting the great potential of paeoniflorin as a drug for preventing PCV2. When the concentration and treatment method of PF were the same, the inhibition rate of the PCV2 *CAP* gene expression was the highest at 48 h (Figure 3E), and the Western blot results were consistent with the quantitative results (Figure 3F and Appendix A). Samples were collected after the specified treatment time and subjected to IFA experiments. The results showed that the fluorescence expression in the PF-treated group was significantly lower than that in the negative control group (Figure 3G). Therefore, PF has a significant anti-PCV2 replication effect in PK15 cells.

### 3.4. Transcriptome Analysis of PK15 Cells with PF Treatment and PCV2 Infection

To explore potential regulators associated with the cellular responses to PF (200 mM) treatment, cell samples treated by PF for 48 h and corresponding controls were collected for transcriptome analysis with the RNA-seq technique. After quality control, an average of 48.9 million clean reads per sample was obtained (Appendix A). Differential expression analysis was then performed to identify transcriptomic differences between the two groups. The principal component analysis plot showed a high degree of similarity between the samples and a high degree of reliability in the sequencing results (Figure 4A). Hierarchical clustering analysis was then performed based on expression levels to show the differences in the expression patterns of differentially expressed genes between the two groups. Differential gene grouping cluster plots showed significant differences between the blank-treated group and the PF-treated group (Figure 4B). Differential gene expression analysis showed that a total of 1787 differentially expressed genes were detected, with 742 differentially upregulated and 1045 differentially downregulated genes (Figure 4C and Appendix A). KEGG enrichment analysis of all the differentially expressed genes in the combined analysis showed that DEGs were mainly enriched in cytokine–cytokine receptor interactions and the PI3K/AKT signaling pathway (Figure 4D).

We then used the previous sequencing results of the subject group [26] and jointly analyzed the negative control group, the PF-treated group, and the PCV2 infection group. Differential expression analysis was then performed to identify transcriptomic differences between the two groups. We found that a total of 455 genes co-increased, 489 genes co-decreased, and 31 genes showed different trends in the comparison between the two groups. In addition, the results showed that gene expression underwent similar changes in these two groups. (Figure 5A and Appendix A). The KEGG enrichment analysis of all the differentially expressed genes in the combined analysis showed that DEGs were mainly enriched in cytokine–cytokine receptor interactions, the PI3K/AKT signaling pathway, and the MAPK signaling pathway (Figure 5B and Appendix A). After reading the relevant literature and conducting preliminary experiments, we selected the PI3K/AKT signaling pathway, which ranks second in KEGG, to complete the subsequent verification and further experiments.

### 3.5. PF Modulates the AKT/mTOR Pathway to Inhibit Autophagy to Suppress PCV2 Replication

To test whether PF (200 mM) can regulate the AKT/mTOR pathway, we examined the activity of the AKT/mTOR pathway by Western blot and performed a rescue experiment utilizing the autophagy activator rapamycin. The results indicated that the addition of PF probably contributed to the activity enhancement of the AKT/mTOR pathway, while RAPA effectively inhibited this effect (Figure 6A,B and Appendix A). However, the addition of RAPA diminished the efficacy of PF, resulting in a rescue of PCV2’s inhibitory effects (Figure 6C,D and Appendix A). To further validate the results regarding the activity of the AKT/mTOR pathway, we conducted a detection of the autophagy level downstream of this pathway. The samples were divided into four groups: the blank control group, the PF-treated group, the autophagy activator RAPA-treated group, and the group treated with both PF and rapamycin. Western blot was employed to determine the expression levels of the P62 and LC3 proteins. The results demonstrated that in the PF-treated group, the expression of the LC3-II protein was significantly reduced. Conversely, in the rapamycin-treated group, the expression of the LC3-II protein was elevated. When cells were treated with both PF and rapamycin, the effect of PF on decreasing the expression of the LC3-II protein was diminished (Figure 6E and Appendix A). For the purpose of obtaining a more comprehensive understanding, we further explored the impact of PCV2 infection on the level of cellular autophagy. Specifically, we utilized Western blot to analyze the expression levels of P62 and LC3 proteins in four distinct groups: the blank infection group, the PF-treated group after infection, the autophagy activator RAPA-treated group after infection, and the group co-treated with PF and RAPA. The results demonstrated that after PCV2 infection, the expression of the LC3-II protein was significantly enhanced. However, when PF treatment was applied, the expression of the LC3-II protein was markedly reduced. When RAPA was added to the PCV2-infected cells, the increase in LC3-II protein expression was further aggravated. When PCV2-infected cells were treated with both PF and RAPA simultaneously, the inhibitory effect of PF on the expression of the LC3-II protein was neutralized by RAPA (Figure 6F and Appendix A).

To further validate the aforementioned results, we opted to monitor autophagy flux using the lentiviral LV-mRFP-GFP-LC3 vector. It should be noted that red LC3 spots represent autophagosomes and autolysosomes, while green LC3 spots only represent autophagosomes. The obtained images presented a clear pattern. PF treatment led to a decrease in yellow and red spots. Conversely, autophagy activator RAPA treatment resulted in an increase in yellow and red spots. Notably, when these two treatments were combined, the reduction in yellow and red spots was effectively attenuated (Figure 7 and Figure 8). In conclusion, these results strongly suggest that PF inhibits the production of intracellular autophagy by promoting the phosphorylation of the AKT/mTOR pathway. This finding not only deepens our understanding of the underlying mechanisms but also paves the way for further research and potential therapeutic applications in the context of PCV2 infection and autophagy regulation.

In order to verify the above results, we further conducted experiments on PF (200 mM) and autophagy activator RAPA-treated groups and detected the expression level of PCV2 *CAP* by Western blot and qPCR. The results indicated that PF had a significant inhibitory effect on PCV2. When we added the autophagy activator RAPA to the treatment, the results showed that the replication of the virus was promoted when RAPA was added alone, and the inhibitory effect of PF on the virus was attenuated when PF and RAPA were added together. It can be seen that RAPA counteracted the inhibitory effect of PF on virus replication. Taking the above results together, we can clearly speculate that PF reduces the replication of the virus through the inhibition of autophagy (Figure 9A). Meanwhile, the results of indirect immunofluorescence and Western blot also proved this (Figure 9B,C).

## 4. Discussion

In this research, we investigated the antiviral impact of paeoniflorin on PK15 cells infected with PCV2. In previous studies, paeoniflorin has been shown to exert an inhibitory effect on several human and mammalian viruses, which is assumed to be a broadly active antiviral plant extract [27,28]. The results indicated a dose-dependent inhibitory effect, highlighting a significant relationship between paeoniflorin and PCV2 infection. Upon treating PCV2-infected cells with paeoniflorin, alterations in the AKT/mTOR signaling pathway were observed, implying that this compound may target this pathway. Additional experimental results revealed that the inhibition of the AKT/mTOR pathway by paeoniflorin promotes autophagy, which, in turn, curtails the replication and spread of PCV2 in PK15 cells. These findings underscore the therapeutic potential of paeoniflorin in managing PCV2 infections post-onset and suggest its promising role as a therapeutic agent against PCV2.

To clarify the mechanism by which PF inhibits PCV2 replication, it can be stated more accurately as follows: When PF enters the cell, it targets the AKT/mTOR pathway and promotes this pathway. This promotion leads to an inhibition of autophagy. Since autophagy has been shown to be a critical process that supports PCV2 replication, the enhancement of autophagy by the inhibition of the AKT/mTOR pathway actually works to suppress PCV2 replication. Thus, PF effectively reduces PCV2 replication by inhibiting autophagy through the promotion of the AKT/mTOR pathway (Figure 10).

Moreover, there exists a close connection between the AKT/mTOR pathway and viral replication. The AKT/mTOR pathway is a significant signaling pathway that participates in the regulation of cell growth, proliferation, and metabolism [29]. When a virus infects a cell, it utilizes the host cell’s signaling pathways to create favorable conditions for its own replication and survival [30]. The AKT/mTOR pathway has a crucial regulatory function in viral replication. Some viruses can activate this pathway to enhance their replication. For instance, by activating the AKT/mTOR pathway, viruses like the hepatitis B virus (HBV) and human immunodeficiency virus (HIV) can modify the metabolic state of the host cells, thereby providing sufficient energy and raw materials to support viral replication [31]. During the infection with the hepatitis B virus (HBV), the activation of the PI3K/Akt/mTOR signaling pathway is capable of elevating the viral load and the extent of liver fibrosis [32]. In the context of human immunodeficiency virus (HIV) infection, the PI3K/Akt signaling pathway assumes a critical role. It regulates T-cell and mitochondrial functions, promotes HIV replication, re-activates latent HIV, and sustains the viral reservoir [33]. On the contrary, other viruses, such as the influenza A virus and adenovirus, choose to inhibit the AKT/mTOR pathway to take advantage of the host cell’s antiviral mechanisms [34]. Adenoviruses utilize the E1A protein to bind to transcriptional co-activators such as p300/CBP. This binding impedes the interaction between these co-activators and the transcription factors associated with the AKT/mTOR pathway. Meanwhile, the E1A protein down-regulates the expression or activity of PI3K, reducing AKT phosphorylation and thereby suppressing the AKT/mTOR pathway [35]. The inhibition of the AKT/mTOR pathway activates cellular autophagy. Adenoviruses take advantage of this to induce moderate autophagy, supplying essential nutrients like amino acids for viral replication and assembly. Moreover, autophagosomes can serve as sites for viral replication, enabling the virus to evade recognition and clearance by the host immune system [35]. Influenza A viruses make use of the interaction between the NS1 protein and the regulatory subunit p85 of PI3K. This prevents the interaction between PI3K and AKT, inhibits AKT phosphorylation and activation, and further suppresses the AKT/mTOR pathway, thus interfering with the host cell’s immune response [36,37]. In doing so, these viruses can restrict the growth and proliferation of host cells and maintain their own stable state, thus evading the host’s immune attack. In this study, we have shown that the inhibitory effect of paeoniflorin on PCV2 is achieved through activating the AKT/mTOR signaling pathway. It is implied that the PCV2 infection inhibits the AKT/mTOR pathway to utilize the host cell’s antiviral mechanisms to assist its own replication. Paeoniflorin, however, can attenuate this inhibition, activating the AKT/mTOR pathway to regulate the replication of PCV2.

Appropriate autophagy is a necessary condition for the host’s defense and immune responses [38]. Specifically, autophagy can serve as an effective defense mechanism against various viruses by degrading viral components or particles [38,39]. Meanwhile, numerous studies have demonstrated that multiple viruses can induce autophagy to facilitate their replication [40]. These findings indicate that autophagy has a dual role in the host’s antiviral responses. There is growing evidence suggesting that PCV2 infection can trigger autophagic processes in both PK-15 cells and 3D4/21 cells, which promotes PCV2 replication through different mechanisms [41,42]. Consequently, it is of great significance to investigate the mechanism of action of PF in inhibiting PCV2. In this study, it was found that not only is the AKT/mTOR signaling pathway closely related to the inhibition of PCV2 replication, but autophagy, which is downstream of the AKT/mTOR signaling pathway, also plays an important role [43,44]. The research results show that PF reduces the generation of intracellular autophagy by inhibiting the autophagy induced by PCV2, thereby reducing the replication of PCV2, which takes advantage of autophagy. To conclude, the results of this study show that PF significantly inhibits PCV2 replication in PK15 cells. Specifically, PF inhibits the intracellular autophagy induced by PCV2 by activating the AKT/mTOR signaling pathway to suppress virus replication. These research findings suggest that PF could be a promising and effective drug for preventing and treating PCV2 infections.

Although this study has obtained certain achievements in investigating the mechanism of PF against PCV2 infection, it is nonetheless accompanied by several limitations. The current research is principally restricted to the cellular level, which implies a lack of comprehensive verification within a whole animal model. Consequently, this shortcoming renders it extremely challenging to conduct a thorough assessment of the therapeutic efficacy and potential side effects of PF in intricate in vivo environments during PCV2 infection. Furthermore, the detailed molecular mechanisms through which PF activates the AKT/mTOR signaling pathway suppresses autophagy and, subsequently, the impact of PCV2 replication remains inadequately elucidated. In particular, the alterations in the modification of key proteins within the pathway and their interactions with viral proteins demand further in-depth exploration.

In future studies, it is of the utmost importance to establish appropriate animal models. In doing so, a comprehensive evaluation of the pharmacokinetic, pharmacodynamic properties, and safety of PF in vivo can be carried out. This evaluation can, in turn, ensure a more dependable foundation for its clinical application. Additionally, a meticulous analysis of the precise mechanism by which PF acts on the AKT/mTOR signaling pathway and autophagy-related molecules is indispensable [45]. Such an analysis will permit accurate targeting and regulation of virus replication. Concurrently, delving into and capitalizing on the potential of paeoniflorin as an antiviral agent is expected to initiate new trials and introduce novel and efficient strategies for the prevention and treatment of PCV2 infectious diseases.

## 5. Conclusions

In summary, this study shows that paeoniflorin significantly enhances the resistance to PCV2 infection in PK15 cells. In addition, paeoniflorin may increase inhibited PCV2 replication by activating AKT/mTOR signaling. It regulates downstream target proteins through the AKT/mTOR signaling pathway to inhibit the production of intracellular autophagy. These findings indicate that paeoniflorin may be suitable as an effective therapeutic agent for targeting anti-PCV2 infection.

## Figures and Tables

**Figure 1 vetsci-12-00117-f001:**
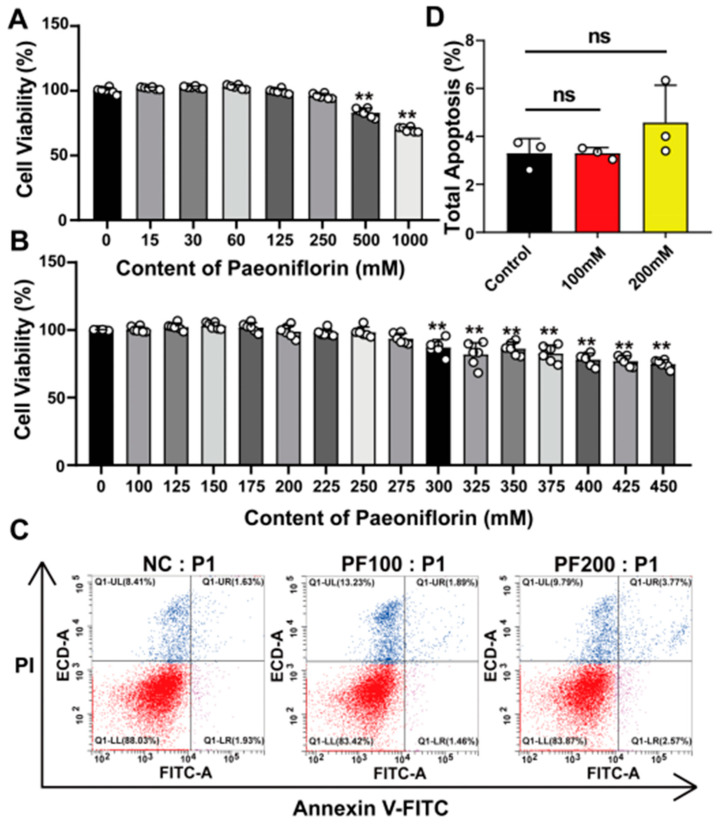
The safety assessment of PF. (**A**) The gradient dilution method was used to determine the approximate working concentration of PF. (**B**) The effect of 0–450 mM PF on the activity of PK15 cells was detected through the CCK8 method. (**C**) The apoptosis levels in cells treated with 100 mM and 200 mM PF were detected by flow cytometry. (**D**) Statistical analysis of the flow cytometry was performed to detect the apoptosis levels in PK15 cells after PF treatment. All data are presented as the mean ± SD, ^ns^
*p* > 0.05, ** *p* < 0.01.

**Figure 2 vetsci-12-00117-f002:**
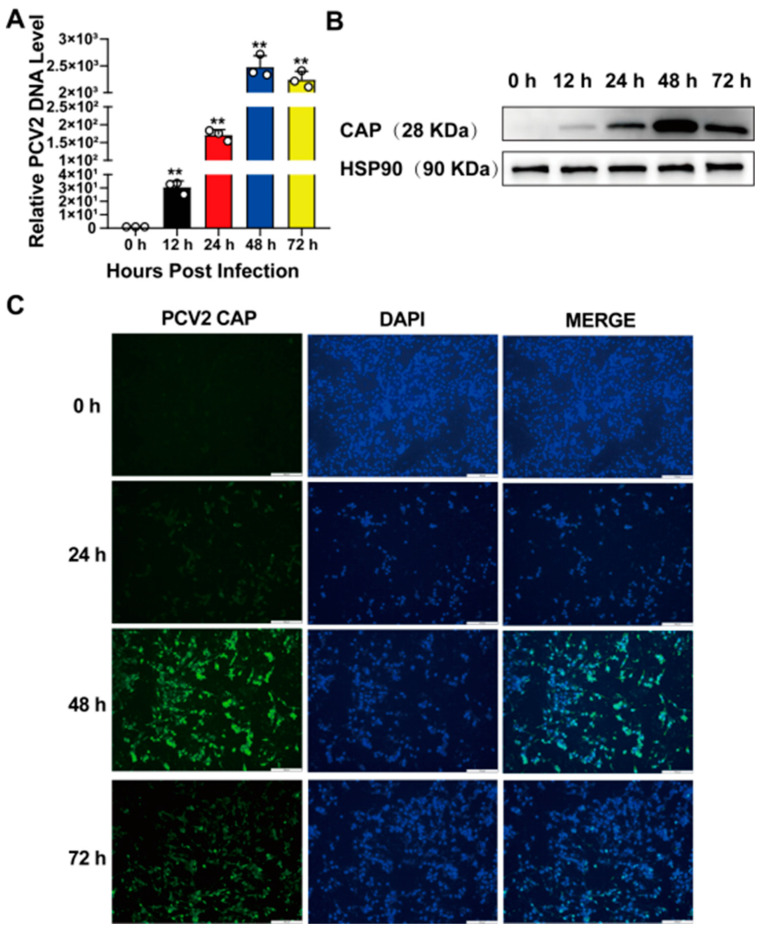
The establishment of the PCV2 infection model. (**A**) PK15 cells were infected with PCV2 for designated lengths of time (0, 12, 24, 48, 72 h), and the relative expression of PCV2 CAP was measured using qPCR. (**B**) The protein expression of the PCV2 capsid protein in PK15 cells within the specified time periods was analyzed. (**C**) The results of the fluorescent expression of the PCV2 capsid protein in PK15 cells within the specified time periods are shown. The image scale is 100 μm. All data are presented as the mean ± SD, ** *p* < 0.01.

**Figure 3 vetsci-12-00117-f003:**
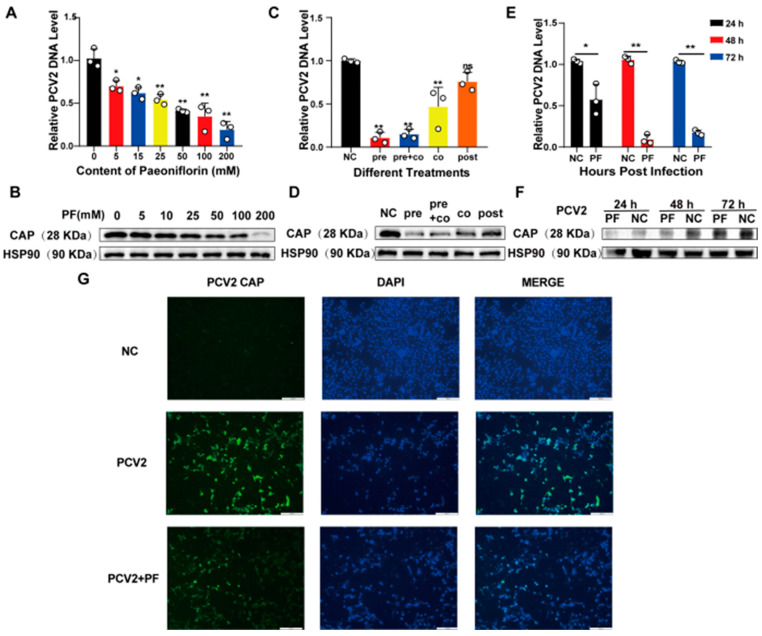
The effect of PF on PCV2 replication. (**A**,**B**) Different concentrations of PF (0 mM, 5 mM, 15 mM, 25 mM, 50 mM, 100 mM, 200 mM) were used to treat PCV2-infected cells. Subsequently, the DNA expression level of PCV2 CAP was detected by qPCR, and the expression levels of PCV2 CAP were determined by Western blot. (**C**,**D**) PCV2-infected cells were treated with different PF addition methods (pre: adding PF 24 h before adding PCV2; co: adding PF simultaneously with adding PCV2; post: adding PF 24 h after adding PCV2). Then, the DNA expression level of PCV2 CAP was detected by qPCR, and the expression levels of PCV2 CAP were detected by Western blot. (**E**,**F**) PCV2-infected cells were treated with PF for the indicated times. After that, the DNA expression level of PCV2 CAP was detected by qPCR, and the expression levels of PCV2 CAP were detected by Western blot. (**G**) The expression of PCV2 CAP on PK15 cells after 48 h of pre-treatment of PF and the protein fluorescence expression on PK15 cells are presented. The scale of the image is 100 μm. All data are presented as the mean ± SD, ^ns^
*p* > 0.05, * *p* < 0.05, ** *p* < 0.01.

**Figure 4 vetsci-12-00117-f004:**
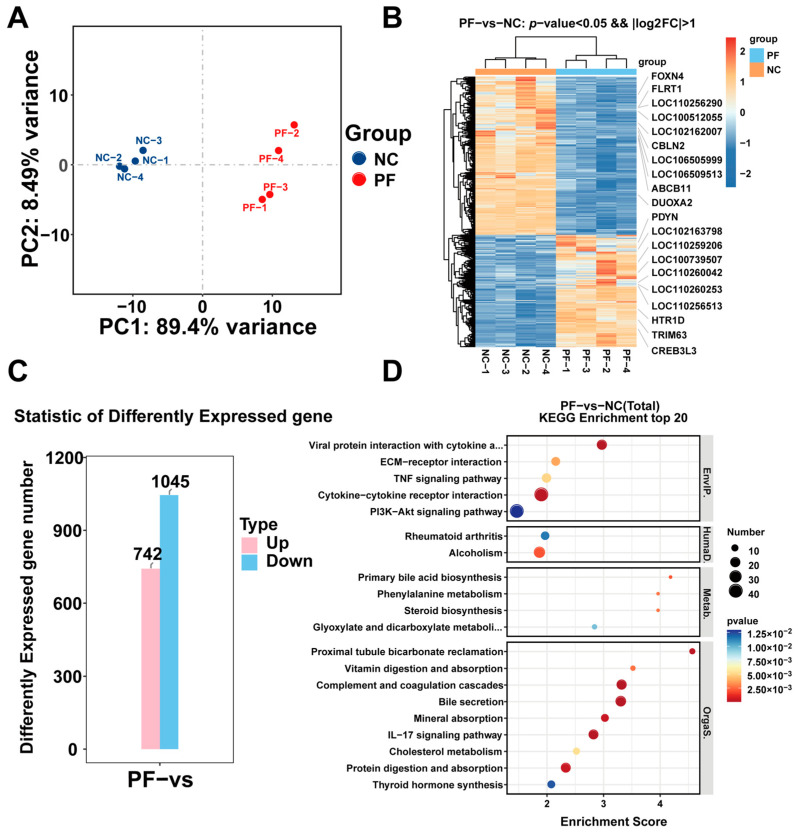
Sequencing results of PF-treated (200 mM) and negative control groups. (**A**) Principal component analysis plot showing the similarity between sequencing groups. (**B**) Clustering plot of differential gene grouping, where red indicates the relatively high expression of protein-coding genes and blue indicates the relatively low expression of protein-coding genes. (**C**) Statistical histogram of differentially expressed genes showing the number of differentially expressed genes that were up-regulated and down-regulated. (**D**) KEGG bubble map: the bubbles represent the top 20 KEGG pathways enriched in DEGs. The size of the dots indicates the degree of DEG enrichment. The color of the dots indicates the importance of DEG enrichment.

**Figure 5 vetsci-12-00117-f005:**
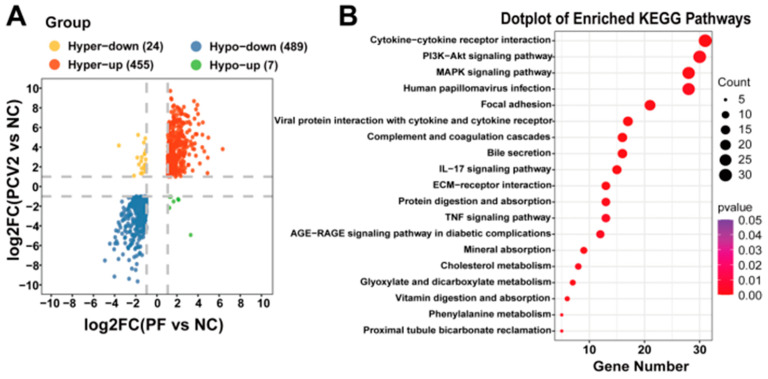
Sequencing results of PF-treated, PCV2-infected, and negative control groups. (**A**) The four-quadrant plot of differential genes, in which red denotes co-increasing differential genes; blue denotes co-decreasing differential genes; yellow denotes differential genes that rise in PCV2 infection sequencing and fall in PF-treated sequencing; and green denotes differential genes that fall in PCV2 infection sequencing and rise in PF-treated sequencing rising differential genes. (**B**) KEGG bubble map: the bubbles represent the top 20 KEGG pathways enriched in DEGs. The size of the dots indicates the degree of DEG enrichment. The color of the dots indicates the importance of DEG enrichment.

**Figure 6 vetsci-12-00117-f006:**
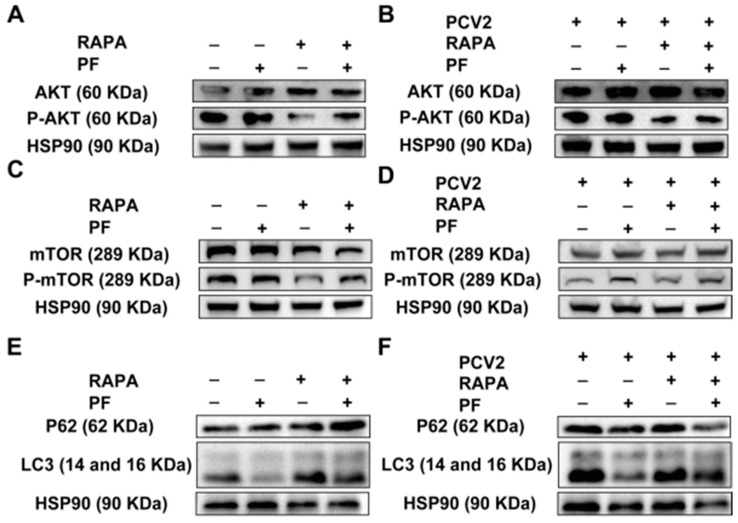
Effect of PF (200 mM) on the AKT/mTOR pathway and its downstream autophagy (**A**,**B**) After treating PK15 cells with PF and RAPA for the indicated durations, the activity of the AKT and mTOR pathways was measured. (**C**,**D**) After treating PCV2-infected cells with PF and RAPA for the indicated durations, the activity of the AKT and mTOR pathways was measured. (**E**) After treating PK15 cells with PF and RAPA for the indicated durations, the levels of LC3 and P62 were detected. (**F**) After treating PCV2-infected cells with PF and RAPA for the indicated durations, the levels of LC3 and P62 were detected.

**Figure 7 vetsci-12-00117-f007:**
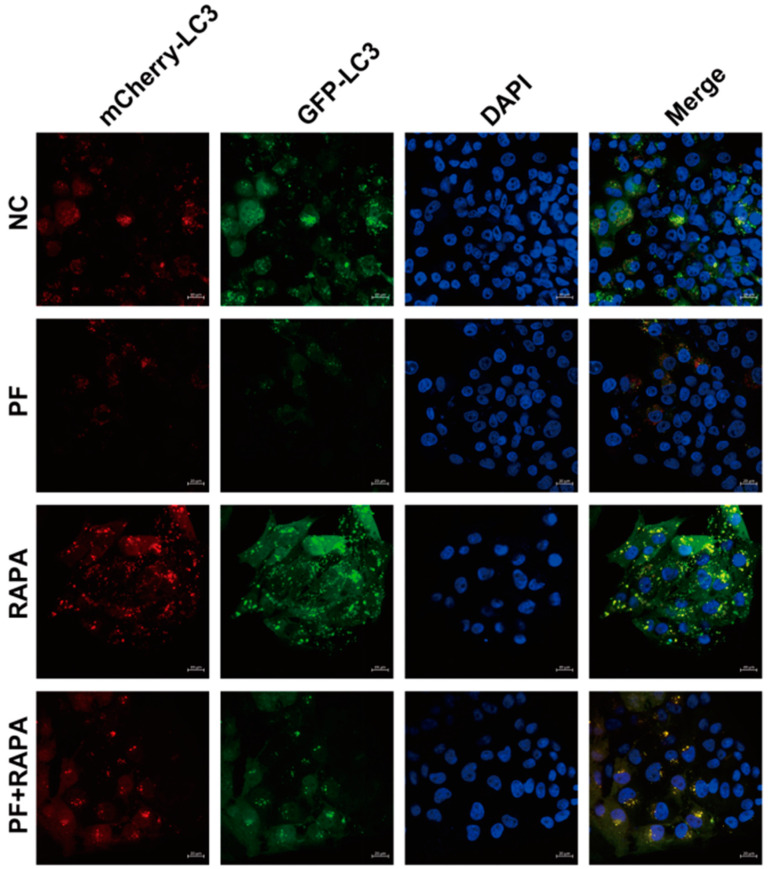
After PK15 cells were treated with PF (200 mM) and RAPA for the indicated times, confocal fluorescent images of PK15 cells expressing mRFP-GFP-LC3 were obtained. In the confocal fluorescence image, red LC3 dots represent autophagosomes and autolysosomes, green LC3 dots represent only autophagosomes and blue dots represent the nucleus. The scale of the image is 20 μm.

**Figure 8 vetsci-12-00117-f008:**
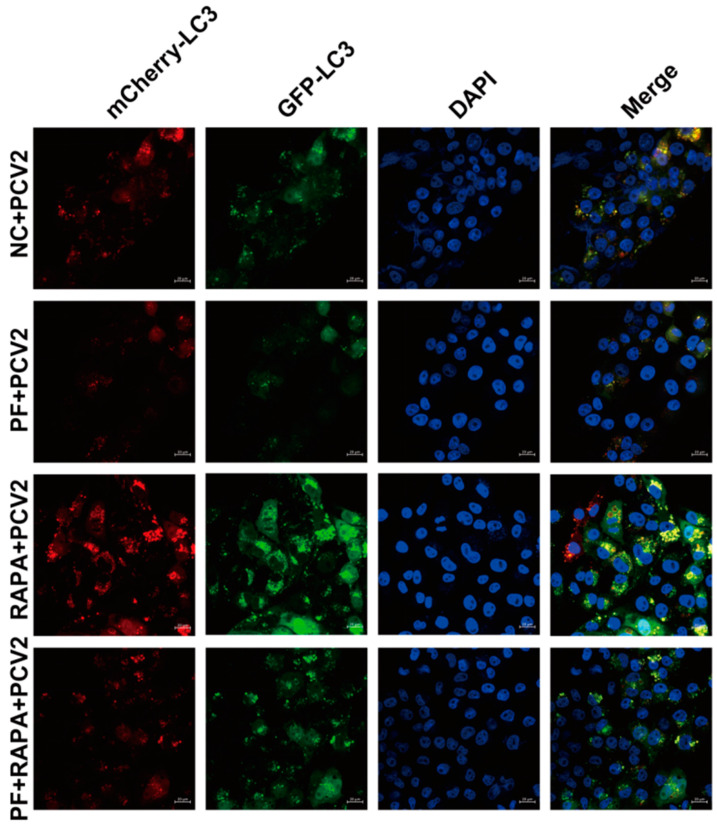
After PCV2-infected cells were treated with PF (200 mM) and RAPA for the indicated times, confocal fluorescent images of PK15 cells expressing mRFP-GFP-LC3 were obtained. In the confocal fluorescence images, red LC3 spots signify autophagosomes and autolysosomes, green LC3 spots indicate only autophagosomes and blue spots represent the nucleus. The scale of the image is 20 μm.

**Figure 9 vetsci-12-00117-f009:**
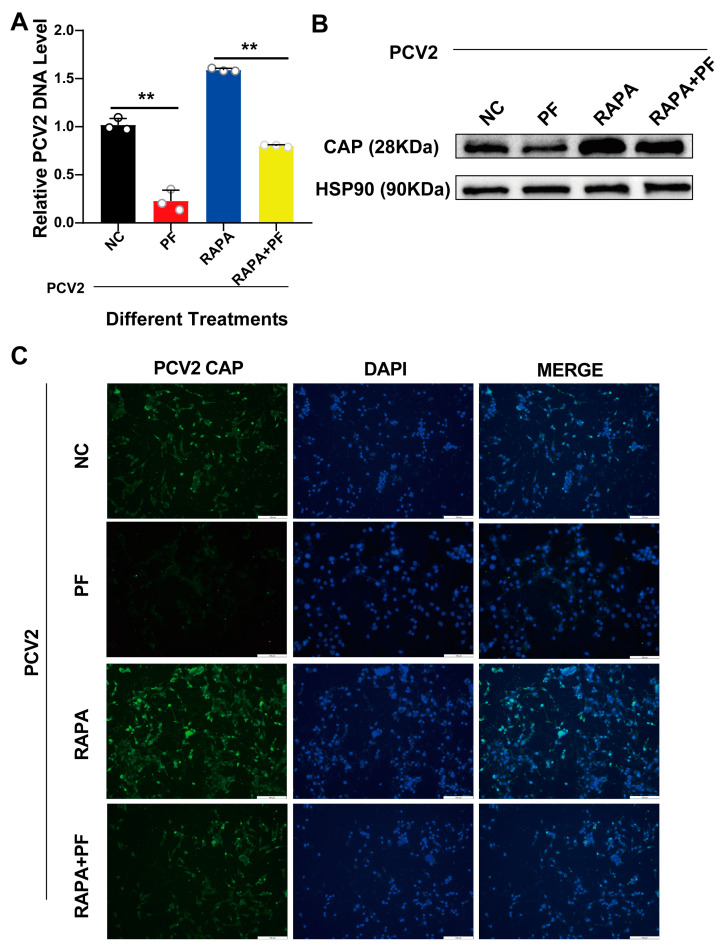
The effect of PF (200 mM) on PCV2 replication through autophagy and a rescue experiment with the autophagy activator RAPA. (**A**) The copy number of porcine circovirus type 2 (PCV2) in PK15 cells was counted after the treatment with PF and RAPA. (**B**) The protein expression of the PCV2 capsid protein (CAP) in PK15 cells was examined after PF and RAPA treatment. (**C**) The results of the fluorescent expression of the PCV2 capsid protein in PK15 cells were obtained after the application of PF and RAPA treatment. All data are presented as the mean ± SD, ** *p* < 0.01.

**Figure 10 vetsci-12-00117-f010:**
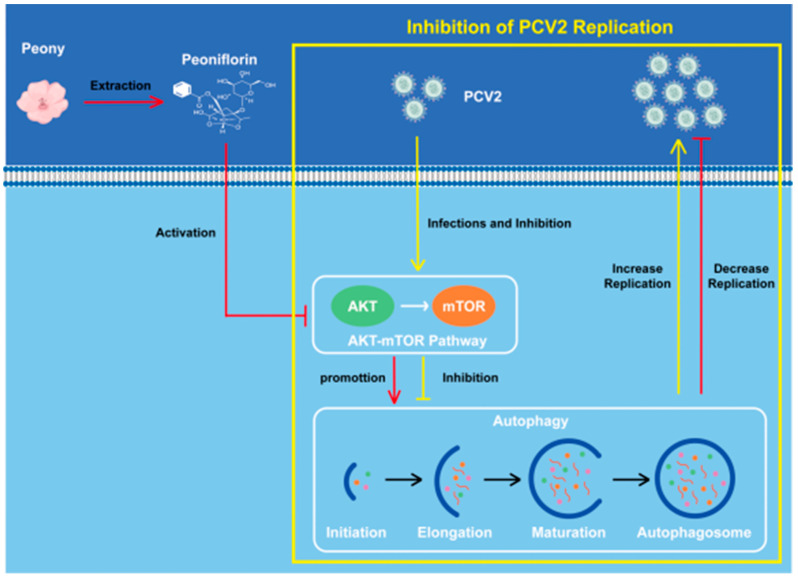
Mechanism of PF inhibiting PCV2 replication by influencing autophagy through the AKT/mTOR pathway.

## Data Availability

The RNA-seq data have been submitted to NCBI′s SRA repository under BioProject IDs: PRJNA1173311.

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
