# Peer review of "Paeoniflorin Inhibits Porcine Circovirus Type 2 Replication by Inhibiting Autophagy and Targeting AKT/mTOR Signaling"

_vetsci, 2025, doi:10.3390/vetsci12020117_

Round 1
Reviewer 1 Report
Comments and Suggestions for Authors
The following work describes the role of an extract in inhibiting PCV-2 infection in vitro via its action on autophagy. The work presents originality, methodological rigor, and is of good quality. Below are some of my comments to improve the readability of the manuscript and to definitively validate the results obtained and described.
1) Abstract: The abstract should be more specific about the methods used and how some results were obtained.
2) Introduction: The introduction explains many mechanisms related to the pathogenicity of PCV but lacks some essential basic information (e.g., classification, description of the virion, worldwide diffusion, role of the wild boar, etc.). Below are some references to improve the first part of the introduction: Porcine circoviruses: a review (for general information); Porcine circoviruses: current status, knowledge gaps, and challenges (for current spread and other information); Detection of selected pathogens in reproductive tissues of wild boars in the Campania region, southern Italy (for reproductive tract colonization in wild boar).
3) CCK-8 Assay Line 117: How were the doses chosen?
4) Flow cytometry: What about compensation?
5) Are a whole membranes of the figures available? Can you make supplementary files?
6) Discussion: The authors should discuss other roles of PI3K/Akt/mTOR in other viruses (e.g., entry, initiation of apoptosis, etc.). Furthermore, it should cite other studies that have used the PI3K/Akt/mTOR axis for antiviral therapy (e.g., SARS-CoV-2, Influenza, and FeHV-1 to remain in veterinary viruses): PI3K/Akt/Nrf2-mediated cellular signaling and virus-host interactions: latest updates on the potential therapeutic management of SARS-CoV-2 infection (SARS-CoV-2); Inhibition of Influenza A Virus Replication by Antagonism of a PI3K-AKT-mTOR Pathway Member Identified by Gene-Trap Insertional Mutagenesis (Influenza); Antiviral Activity of Nitazoxanide and Miltefosine Against FeHV-1 In Vitro AND Modifications of the PI3K/Akt/mTOR Axis during FeHV-1 Infection in permissive cells (for FHV-1).
Author Response
Review 1
The following work describes the role of an extract in inhibiting PCV-2 infection in vitro via its action on autophagy. The work presents originality, methodological rigor, and is of good quality. Below are some of my comments to improve the readability of the manuscript and to definitively validate the results obtained and described.
- Abstract: The abstract should be more specific about the methods used and how some results were obtained.
Answer: Thanks for your valuable advice. I have made the modifications in Abstract section.
2) Introduction: The introduction explains many mechanisms related to the pathogenicity of PCV but lacks some essential basic information (e.g., classification, description of the virion, worldwide diffusion, role of the wild boar, etc.). Below are some references to improve the first part of the introduction: Porcine circoviruses: a review (for general information); Porcine circoviruses: current status, knowledge gaps, and challenges (for current spread and other information); Detection of selected pathogens in reproductive tissues of wild boars in the Campania region, southern Italy (for reproductive tract colonization in wild boar).
Answer: Thanks for your valuable advice. I have made the modifications in Introduction section (Line 40-46).
3) CCK-8 Assay Line 117: How were the doses chosen?
Answer: Thanks for your valuable advice. First, I determined that the unit of the dosage of paeoniflorin was mM according to the information on the MCE website. Then, in Figure 1A, through the semi-logarithmic dilution method, it was found that the safe concentration range was 0-500 mM. Subsequently, in Figure 1B, the concentration intervals were narrowed to set different concentrations, aiming to obtain a more specific safe concentration range.
4) Flow cytometry: What about compensation?
Answer: Thanks for your valuable advice. About Flow cytometry, our results were shown in the figure below:
5) Are a whole membranes of the figures available? Can you make supplementary files?
Answer: Thanks for your valuable advice. In fact, according to Veterinary Sciences requirement, we have uploaded the original Images for BlotsGels in the process of submission.
6) Discussion: The authors should discuss other roles of PI3K/Akt/mTOR in other viruses (e.g., entry, initiation of apoptosis, etc.). Furthermore, it should cite other studies that have used the PI3K/Akt/mTOR axis for antiviral therapy (e.g., SARS-CoV-2, Influenza, and FeHV-1 to remain in veterinary viruses): PI3K/Akt/Nrf2-mediated cellular signaling and virus-host interactions: latest updates on the potential therapeutic management of SARS-CoV-2 infection (SARS-CoV-2); Inhibition of Influenza A Virus Replication by Antagonism of a PI3K-AKT-mTOR Pathway Member Identified by Gene-Trap Insertional Mutagenesis (Influenza); Antiviral Activity of Nitazoxanide and Miltefosine Against FeHV-1 In Vitro AND Modifications of the PI3K/Akt/mTOR Axis during FeHV-1 Infection in permissive cells (for FHV-1).
Answer: Thanks for your valuable advice. I have made revisions according to the suggestions and provided examples of specific viruses in the text to demonstrate the research on the PI3K/Akt/mTOR axis in relation to viruses (Line 460-482).

Reviewer 2 Report
Comments and Suggestions for Authors
In this manuscript, the authors explore the inhibitory effect of Paeoniflorin (PF) on porcine circovirus type 2 (PCV2) replication in PK-15 cells and its potential mechanisms. They revealed for the first time that PF inhibits autophagy by targeting the AKT/mTOR signaling pathway, thereby inhibiting PCV2 replication, providing a potential strategy for the treatment of porcine PCV2 infection. These findings are not only innovative but fascinating. However, there are still some issues in the study of the mechanism of the effect of PF on PCV2.
Major concerns:
1. In the section "3.3. Effect Analysis of PF on PCV2 Replication", there is no positive control drug for PCV used.
2. In Figures 6A and, 6B, the addition of PF did not increase the expression levels of P-ATK and P-mTOR proteins, and therefore it cannot be concluded that "the addition of PF significantly enhanced the activity of the AKT/mTOR pathway".
3. In Figures 6C and 6D, the sample groups were all infected with PCV2, and there were no uninfected controls. Therefore, the authors could not conclude that "PCV2 infection inhibits the activity of the AKT/mTOR pathway".
4. In Figures 6E and 6F, why no change in P62 protein expression when cells are treated with PF and RAPA? Please explain this phenomenon and cite the appropriate references.
5. To distinguish the infected group from the uninfected group, please mark infected PCV2 in Figure 9.
6. The mechanism diagram is vague and suggested to be corrected.
7. Why did the authors verify the effect of PF on AKT-mTOR-autophagy? What is the basis? The mTOR-autophagy pathway does not seem to be observed in the transcriptomic data. Please clarify.
Minor comments:
1. Why is the result of the 50 mM BF treatment included in the text and figure legends, but not observed in the Figure 1?
2. The legends for Figures 1C and 1D are confusing, please check.
3. Whether the qPCR results in Figure 2A should be significance analysed.
4. Please label the protein size in all western blot images.
5. Please explain clearly the concentration of PF used in the manuscript/images.
6. It is suggested to merge the images of one result into one figure.
7. Please improve the readability of the manuscript.
Author Response
Review 2
In this manuscript, the authors explore the inhibitory effect of Paeoniflorin (PF) on porcine circovirus type 2 (PCV2) replication in PK-15 cells and its potential mechanisms. They revealed for the first time that PF inhibits autophagy by targeting the AKT/mTOR signaling pathway, thereby inhibiting PCV2 replication, providing a potential strategy for the treatment of porcine PCV2 infection. These findings are not only innovative but fascinating. However, there are still some issues in the study of the mechanism of the effect of PF on PCV2.
Major concerns:
- In the section "3.3. Effect Analysis of PF on PCV2 Replication", there is no positive control drug for PCV used.
Answer: Thanks for your valuable advice. I didn't take the use of positive drug control into account in my initial experimental design. I'm grateful for the expert's advice and will pay attention to this aspect in future experiments. However, referring to the research by Yang Hu et al. in 2022, paeoniflorin had the most obvious anti-White Spot Syndrome Virus (WSSV) effect on shrimp infected with WSSV, with a maximum protection efficiency of more than 60%. This provides a theoretical basis for the experiment.
- In Figures 6A and, 6B, the addition of PF did not increase the expression levels of P-ATK and P-mTOR proteins, and therefore it cannot be concluded that "the addition of PF significantly enhanced the activity of the AKT/mTOR pathway".
Answer: Thanks for your valuable advice. I'm grateful to the expert for the suggestions. The view that "the addition of PF significantly enhanced the activity of the AKT/mTOR pathway" was derived from multiple repeated experiments and calculation of the gray values in Western blot, so it should be relatively reliable. Thus, we also revised the conclusion “the addition of PF probably contributes to the activity enhancement of the AKT/mTOR pathway” (Line 354).
- In Figures 6C and 6D, the sample groups were all infected with PCV2, and there were no uninfected controls. Therefore, the authors could not conclude that "PCV2 infection inhibits the activity of the AKT/mTOR pathway".
Answer: Thanks for your valuable advice. The relevant impacts on the pathway before and after viral infection have been reported in the article "miR-214-5p/C1QTNF1 axis enhances PCV2 replication through promoting autophagy by targeting AKT/mTOR signaling pathway". In Figures 6C and 6D, the addition of RAPA diminishes the efficacy of PF, resulting in a rescue of PCV2's inhibitory effects. Besides, we also deleted the description of conclusion “We subsequently conducted an infection with PCV2, the results showed that PCV2 infection inhibited the activity of the AKT/mTOR pathway, on the contrary, PF treatment significantly promoted the activity of the AKT/mTOR pathway, and that PF treatment was able to alleviate the inhibitory effect of PCV2 infection on the pathway when the two acted together at the same time”.
- In Figures 6E and 6F, why no change in P62 protein expression when cells are treated with PF and RAPA? Please explain this phenomenon and cite the appropriate references.
Answer: Thanks for your valuable advice. Referring to the articles, I think the reason why there is no significant change in p62 may be that autophagy first increases and then decreases, resulting in no significant change in its protein.
- To distinguish the infected group from the uninfected group, please mark infected PCV2 in Figure 9.
Answer: Thanks for your valuable advice. I have made the modifications according to the suggestions.
- The mechanism diagram is vague and suggested to be corrected.
Answer: Thanks for your valuable advice. I have made the modifications according to the suggestions.
- Why did the authors verify the effect of PF on AKT-mTOR-autophagy? What is the basis? The mTOR-autophagy pathway does not seem to be observed in the transcriptomic data. Please clarify.
Answer: Thanks for your valuable advice. Through combined analysis, I selected the PI3K-AKT signaling pathway. Then, by referring to the article (miR-214- 5p/C1QTNF1 axis enhances PCV2 replication through promoting autophagy by targeting AKT/mTOR signaling pathway), I made a connection to the mTOR-autophagy pathway.
Minor comments:
- Why is the result of the 50 mM BF treatment included in the text and figure legends, but not observed in the Figure 1?
Answer: Thanks for your valuable advice. I have made the modifications according to the suggestions. The 50 mM concentration was used in the preliminary experiments. I forgot to make the modification when writing the article.
- The legends for Figures 1C and 1D are confusing, please check.
Answer: Thanks for your valuable advice. I have revised the order of Figure 1C and 1D.
- Whether the qPCR results in Figure 2A should be significance analysed.
Answer: Thanks for your valuable advice. I have added the significance analysis.
- Please label the protein size in all western blot images.
Answer: Thanks for your valuable advice. I have added the protein size in all western blot images.
- Please explain clearly the concentration of PF used in the manuscript/images.
Answer: Thanks for your valuable advice. Based on Figures 2A and 2B, I found that paeoniflorin at a concentration of 200 mM exhibited the best effect. Therefore, 200 mM was adopted for subsequent experiments. I have added dosage notations in several places in the article.
- It is suggested to merge the images of one result into one figure.
Answer: Thanks for your valuable advice. Due to the effect on image resolution, the images presenting one result were divided into two parts in Figures 8 and 9.
- Please improve the readability of the manuscript.
Answer: Thanks for your valuable advice. I have made some modifications.

Reviewer 3 Report
Comments and Suggestions for Authors
In this paper, This study demonstrates that paeoniflorin can significantly enhance the resistance of PK15 cells to PCV2 infection and paeoniflorin may increase the inhibition of PCV2 replication by activating the AKT/mTOR signaling pathway. It has been proved that it regulates downstream target proteins through the AKT/mTOR signaling pathway and inhibits the generation of autophagy within cells. These results of the authors suggest that paeoniflorin may be suitable as an effective therapeutic agent against PCV2 infection. There are some minor errors such as punctuation spelling mistakes in the article that need attention, and the pictures lack quantitative analysis. In addition, the authors should pay attention to the standard use of grammar. Here are some of my suggestions and questions.
1. Line 115, is the density of 2,000 PK15 cells per well too low for the CCK-8 assay?
2. Lines 122, 232, etc., lack Spaces between numbers and letters.
3. FIG. 2A does not indicate the significance of the results in the figure
4. In line 262, the word pretretment is misspelled.
5. FIG. 3 and FIG. 6 lack quantitative analysis of wb graph.
6. Whether bubbles exist in the internal reference strip of F and hsp90 in FIG. 3, and whether the analysis results are credible.
7. The results of immunofluorescence and flow cytometry were not quantified.
8. Some of the references in the article are too old.
Author Response
Review 3
In this paper, This study demonstrates that paeoniflorin can significantly enhance the resistance of PK15 cells to PCV2 infection and paeoniflorin may increase the inhibition of PCV2 replication by activating the AKT/mTOR signaling pathway. It has been proved that it regulates downstream target proteins through the AKT/mTOR signaling pathway and inhibits the generation of autophagy within cells. These results of the authors suggest that paeoniflorin may be suitable as an effective therapeutic agent against PCV2 infection. There are some minor errors such as punctuation spelling mistakes in the article that need attention, and the pictures lack quantitative analysis. In addition, the authors should pay attention to the standard use of grammar. Here are some of my suggestions and questions.
- Line 115, is the density of 2,000 PK15 cells per well too low for the CCK-8 assay?
Answer: Thanks for your valuable advice. Upon inspection, I found that it was an error made during writing. It should be 20,000 cells per well.
- Lines 122, 232, etc., lack Spaces between numbers and letters.
Answer: Thanks for your valuable advice and we have revised it.
- 2A does not indicate the significance of the results in the figure
Answer: Thanks for your valuable advice and we have revised it.
- In line 262, the word pretretment is misspelled.
Answer: Thanks for your valuable advice and we have revised it.
- 3 and FIG. 6 lack quantitative analysis of wb graph.
Answer: Thanks for your valuable advice. I added the quantitative analysis in the supplementary file (Figure S1 and Figure S2).
- Whether bubbles exist in the internal reference strip of F and hsp90 in FIG. 3, and whether the analysis results are credible.
Answer: Thanks for your valuable advice. There was a minor issue during the transfer of the hsp90 band. That area isn't a bubble, and the target band is in good condition.
- The results of immunofluorescence and flow cytometry were not quantified.
Answer: Thanks for your valuable advice. For the results of flow cytometry, I have presented them using the total apoptosis rate in Figure 1D.
- Some of the references in the article are too old.
Answer: Thanks for your valuable advice. I have made the modifications according to the suggestions.

Round 2
Reviewer 2 Report
Comments and Suggestions for Authors
The authors have addressed most of my concerns, but there are still a few minor issues.
1. 1. In Fig. 6, the changes in expression of the proteins in each group are not very obvious, please provide additional gray value analysis graphs of the proteins for evidence.
2. Lines 436-439: Since autophagy has been shown to be a critical process that supports PCV2 replication, the enhancement of autophagy by the inhibition of the AKT/mTOR pathway actually works to suppress PCV2 replication. Is it logical?
3. Please revise the mechanism figure again based on the conclusions of the manuscript (PF activates the AKT-mTOR pathway? PCV2 also?).
Comments on the Quality of English Language
The authors have addressed most of my concerns, but there are still a few minor issues.
1. 1. In Fig. 6, the changes in expression of the proteins in each group are not very obvious, please provide additional gray value analysis graphs of the proteins for evidence.
2. Lines 436-439: Since autophagy has been shown to be a critical process that supports PCV2 replication, the enhancement of autophagy by the inhibition of the AKT/mTOR pathway actually works to suppress PCV2 replication. Is it logical?
3. Please revise the mechanism figure again based on the conclusions of the manuscript (PF activates the AKT-mTOR pathway? PCV2 also?).
Author Response
Review 2
1. In Fig. 6, the changes in expression of the proteins in each group are not very obvious, please provide additional gray value analysis graphs of the proteins for evidence.
Answer: Thanks for the advice provided. I have organized and added the gray - value statistics for Figure 6 in the Supplementary Figures.docx file.
2. Lines 436-439: Since autophagy has been shown to be a critical process that supports PCV2 replication, the enhancement of autophagy by the inhibition of the AKT/mTOR pathway actually works to suppress PCV2 replication. Is it logical?
Answer: Thanks for the advice provided. It's my fault. According to the experimental results, PF promotes the AKT/mTOR pathway and inhibits autophagy, rather than inhibiting the AKT/mTOR pathway to enhance autophagy. This was a mistake in my article writing. Thank you very much for the expert's reminder. I have corrected it.
3. Please revise the mechanism figure again based on the conclusions of the manuscript (PF activates the AKT-mTOR pathway? PCV2 also?).
Answer: Thanks for the advice provided. I have made revisions according to the comments. What I want to express is that PF is involved in the process in which PCV2 inhibits the AKT/mTOR pathway to enhance autophagy for its own replication. During this process, PF promotes the AKT/mTOR pathway and inhibits autophagy, thus reducing viral replication.

Reviewer 3 Report
Comments and Suggestions for Authors
Thank you very much~
Author Response
Thank you very much~
Answer: Thanks for your kind comment.